# High Glucose-Induced Cardiomyocyte Damage Involves Interplay between Endothelin ET-1/ET_A_/ET_B_ Receptor and mTOR Pathway

**DOI:** 10.3390/ijms232213816

**Published:** 2022-11-10

**Authors:** Sudhir Pandey, Corina T. Madreiter-Sokolowski, Supachoke Mangmool, Warisara Parichatikanond

**Affiliations:** 1Department of Pharmacology, Faculty of Pharmacy, Mahidol University, Bangkok 10400, Thailand; 2Molecular Biology and Biochemistry, Gottfried Schatz Research Center, Medical University of Graz, 8010 Graz, Austria; 3Department of Pharmacology, Faculty of Science, Mahidol University, Bangkok 10400, Thailand; 4Centre of Biopharmaceutical Science for Healthy Ageing (BSHA), Faculty of Pharmacy, Mahidol University, Bangkok 10400, Thailand

**Keywords:** high glucose, endothelin-1, ET_A_-R, ET_B_-R, mTOR, mitochondrial dysfunction, oxidative stress

## Abstract

Patients with type two diabetes mellitus (T2DM) are at increased risk for cardiovascular diseases. Impairments of endothelin-1 (ET-1) signaling and mTOR pathway have been implicated in diabetic cardiomyopathies. However, the molecular interplay between the ET-1 and mTOR pathway under high glucose (HG) conditions in H9c2 cardiomyoblasts has not been investigated. We employed MTT assay, qPCR, western blotting, fluorescence assays, and confocal microscopy to assess the oxidative stress and mitochondrial damage under hyperglycemic conditions in H9c2 cells. Our results showed that HG-induced cellular stress leads to a significant decline in cell survival and an impairment in the activation of ET_A_-R/ET_B_-R and the mTOR main components, Raptor and Rictor. These changes induced by HG were accompanied by a reactive oxygen species (ROS) level increase and mitochondrial membrane potential (MMP) loss. In addition, the fragmentation of mitochondria and a decrease in mitochondrial size were observed. However, the inhibition of either ET_A_-R alone by ambrisentan or ET_A_-R/ET_B_-R by bosentan or the partial blockage of the mTOR function by silencing Raptor or Rictor counteracted those adverse effects on the cellular function. Altogether, our findings prove that ET-1 signaling under HG conditions leads to a significant mitochondrial dysfunction involving contributions from the mTOR pathway.

## 1. Introduction

Type two diabetes mellitus (T2DM) is a chronic disease associated with impairments in multiple organs, including the heart, kidney, brain, liver, and eye. Clinical studies revealed that cardiovascular diseases are the major cause for diabetes-related morbidity and mortality [1]. Mitochondrial dysfunction is often associated with an altered metabolism in diabetic hearts and is likely due to free fatty acid-induced lipotoxicity and an uncoupling of oxidative phosphorylation [2,3]. Oxidative stress has been implicated for induction of apoptosis in cardiomyocytes exposed to high glucose (HG) conditions via mitochondrial cytochrome c-activated caspase 3 pathway [4,5]. HG impairs intracellular Ca^2+^ balance, leads to ventricular dilatation, and induces systolic dysfunction, all of which affect the contractile function of cardiomyocytes [6,7]. Previous studies have shown that HG treatment (30 mM) for 24 h leads to a prominent loss of viability in H9c2 cells through the induction of reactive oxygen species (ROS), mitochondrial damage, and apoptosis of cells [8,9].

Endothelin-1 (ET-1) is a vasoconstrictive peptide produced primarily in the endothelium, exerting its effects through binding to the endothelin receptors (ET-Rs), ET_A_-R and ET_B_-R. Both ET-Rs are abundantly expressed in cardiac tissues [10]. ET-1 is implicated in the progression of cardiovascular diseases and metabolic disorders, such as insulin resistance and T2DM. In experimental congestive heart failure, the deletion of ET-Rs has been shown to inhibit cardiac hypertrophy [11,12]. Cardiac ET_A_-R ablation protects against ageing-associated myocardial remodeling and contractile dysfunction through autophagy regulation [13]. Previous studies suggested that ET-1 and ET-R expression were upregulated in the heart of diabetic patients and endothelin receptor antagonists (ERAs), BQ123 and bosentan, protected against diabetes-induced heart injury in human ventricular heart cells [14,15]. It has been demonstrated that bosentan (non-selective ERA) improved the expression of fibrotic genes, including transforming growth factor-β (TGF-β), connective tissue growth factor (CTGF), and collagen I, and prevented diabetic heart fibrosis in streptozotocin (STZ)-induced diabetic mice [16]. STZ-induced diabetic rat hearts had elevated levels of ET-1 peptide, ET-1 mRNA, ET_A_-R and ET_B_-R mRNAs, and therapy with bosentan enhanced cardiac performance [17,18]. Alterations in cellular metabolism might be linked to an upregulation of ET-1. In cases of ET-1-induced oxidative stress, BQ123 (ET_A_-R blocker) or BQ788 (ET_A_-R/ET_B_-R dual blocker) have been shown to attenuate increased superoxide production in the arteries and veins of patients with coronary artery disease [19] and have been linked to a significant increase in the activity of antioxidant defenses, including glutathione, superoxide dismutase (SOD), and catalase [20,21].

Another key player in cardiovascular pathologies is mechanistic target of rapamycin (mTOR), a serine/threonine protein kinase, which is part of the phosphatidylinositol-3 kinase (PI3K)/Akt signaling pathway. HG levels activate mTOR, and its hyperactivation reduces the myocardial tolerance to stress induced by metabolic defects. Chronic administration of rapamycin (mTOR inhibitor) reduced oxidative stress; significantly lowered plasma glucose, insulin, and triglyceride levels; and improved cardiac function in diabetic mice [22]. Previous studies have shown that ET-R expression is regulated via adenosine monophosphate-activated protein kinase (AMPK)/mTOR signaling pathways under various cardiovascular stresses [23,24]. A chronically active mTOR caused a negative feedback loop that inhibited insulin receptor substrate-1 (IRS-1) in T2DM. In mammals, mTOR is expressed by a single gene, but it forms two complexes, mTORC1 and mTORC2, with functionally different methods and outcomes, by binding to certain adaptor proteins. The catalytic mTOR subunit, the regulatory-associated protein of mTOR (Raptor), the mammalian lethal with sec-13 protein 8 (mLST8), the DEP domain containing mTOR-interacting protein (DEPTOR), and the proline-rich Akt substrate 40 kDa (PRAS40) are the main component proteins that make up mTORC1. In contrast to mTORC1, mTORC2 has additional distinct subunits, including the rapamycin-insensitive companion of mTOR (Rictor) and the mammalian stress-activated MAP kinase-interacting protein 1 (mSIN1) [25]. A key stress regulator, known as mTOR, enhances oxidative metabolism, mitochondrial biogenesis, NADPH oxidase, and ROS generation through Akt-mTOR signaling [26]. According to previous studies, mTOR had a critical role in maintaining oxidative balance and was involved in the degradation of nuclear factor erythroid 2-related factor 2 (NRF2) [27]. It has been suggested that angiotensin II (Ang II)-mediated mTOR phosphorylation in neonatal rat cardiomyocytes and mTOR acted as the upstream modulator of NRF2 in Ang II-induced cardiac hypertrophy [28].

Previous studies have shown that the expression of ET-1 is increased in diabetic hearts, leading to mitochondrial damage and myofibril disorganization through the production of ROS [29]. The ET_A_-R antagonist ameliorated the HG-induced mitochondrial damage in cardiomyocytes [30]. Mitochondrial fusion and fission processes are important for the maintenance of normal cellular metabolic activities, including respiration, mitochondrial DNA (mtDNA) distribution, apoptosis, cell survival, or Ca^2+^ signaling. Mitofusin (MFN)-1 and MFN-2 at the outer mitochondrial membrane (OMM) and optic atrophy protein 1 (OPA1) at the inner mitochondrial membrane (IMM) interact to promote mitochondrial fusion. Fission protein 1 (FIS1), mitochondrial fission factor (MFF), and mitochondrial dynamics proteins of 49 and 51 kDa (MiD49/51) are recruited through receptors by dynamin-related protein-1 (DRP1) from the cytosol to the OMM at sites of division marked by endoplasmic reticulum (ER) [31]. The pro-fusion state is prevalent under conditions of enhanced energy need, such as during starvation or acute stress, while the contrary happens when cells are exposed to a surplus nutrition supply, such as in obesity or T2DM. Recent studies have reported that ET-1 increased mitochondrial fission in vascular smooth muscle cells, and mitochondrial fission inhibitors suppressed ET-1-induced vasoconstriction [32]. Although intracellular and extracellular substrates can affect the dynamics of mitochondrial fission and fusion, it is unclear how to precisely control these dynamics. Depending on the context, glucose can either activate or inhibit mitochondrial fission. Post-translational modifications also regulate these processes [33]. HG intensifies ROS production, mitochondrial permeability transition pore (mPTP) opening, fission, and cell death [34,35].

ET-1 and ET-R expression are increased in the diabetic hearts, and ERAs protected against diabetes-induced cardiac injury in human ventricular cells. However, the relationship between HG and ET-R signaling has not been clearly identified. The molecular mechanisms of ET-1, as well as the subtype specificity and signal transduction of ET-R-mediated mitochondrial dysfunctions, have not been elucidated for diabetic cardiomyopathies. This study investigated the molecular interplay between endothelin ET-1 signaling and mTOR under HG conditions in H9c2 cells.

## 2. Results

### 2.1. HG Treatment Upregulated ET-1/ET_A_-R/ET_B_-R/mTOR in H9c2 Cardiomyoblasts

To investigate the protective effects of ERAs against HG-induced cardiac stresses, we first performed a concentration-dependent assessment of glucose to identify a minimal amount of glucose that could induce detectable transcriptional and translational changes without causing undesirable injury to cells. The MTT assay over a range of D-glucose concentrations (vc = vehicle control, 20 mM, 30 mM, 50 mM, 60 mM, and 80 mM) for 24 h revealed that 30 mM HG treatment could lead to significant alterations in cell viability (Figure 1A). We found that exposure to HG induces a state of glucose intolerance, which was significantly attenuated by insulin treatment (Figure 1B). Next, we examined whether HG treatment could alter the transcriptional expression of glucose transporters GLUT1 and GLUT4 in H9c2 cells. The mRNA expression levels of GLUT1 and GLUT4 were upregulated, as compared to untreated cells (Figure 1C). However, the mRNA expressions of mitochondrial biogenesis markers, peroxisome proliferator-activated receptor gamma coactivator-1α (PGC-1α) and mitochondrial transcription factor A (TFAM), were significantly downregulated (Figure 1D). Importantly, the expressions of ET-1/ET_A_-R/ET_B_-R were also significantly increased under HG treatment (Figure 1E). Interestingly, the mRNA and protein expression of mTOR/Raptor/Rictor was also significantly enhanced with HG treatment, in contrast to untreated cells (Figure 1F,G). We tested for the protein expression of survival and apoptotic markers to further confirm the effect of HG treatment in H9c2 cells. Our results showed that the protein expression of p-Akt/Akt and Bcl-2 was significantly reduced, while apoptotic executioner c-caspase-3 was prominently increased (Figure 1H). The above results suggested that exposure to HG induces cellular stress that is accompanied by decreased cell viability and induction of apoptosis. Therefore, we used a 30 mM HG concentration for further experimental treatment in H9c2 cells. Collectively, these results suggested that HG exposure induces a state of cellular stress that leads to an abnormally increased expression of endothelin ET-1/ET_A_/ET_B_ receptors, together with mTOR, which are accompanied by decreased expression of mitochondrial biogenesis and survival markers in H9c2 cells.

### 2.2. Treatment with ERAs or Inhibition of mTOR Protected against HG-Induced Oxidative Damage

Several studies have shown that elevated glucose levels increase ROS generation in cardiomyocytes. Cardiomyocytes use more oxygen as a result of metabolic abnormalities incited by hyperglycemia, owing to increased fatty acid oxidation [36]. Due to the oxidation of fatty acid rather than carbohydrates, there is a higher level of oxidative stress in the heart [37]. Therefore, cardiovascular abnormalities caused by ROS were found to be the main drivers in diabetes patient’s heart-related mortality [38]. In this study, the intracellular ROS and mitochondrial superoxide production was significantly enhanced by HG treatment in H9c2 cells (Figure 2A,B). The ROS production was additionally increased upon pretreatment with ET-1. This increased production of ROS was suppressed by blocking either ET_A_-R with ambrisentan or blocking ET_A_-R and ET_B_-R together using bosentan (Figure 2A–D). Next, we tested whether the suppression of mTOR C1/C2 could mitigate the ROS levels in H9c2 cells. The silencing efficiency of siRNAs was evaluated by western blotting for the protein expression of Raptor and Rictor (Appendix A). Interestingly, the silencing of either Raptor (mTORC1) or Rictor (mTORC2) significantly reduced ROS levels under HG conditions. Notably, not only was total ROS release significantly attenuated but so was mitochondrial superoxide production (Figure 2A–D). These results show that the suppression of mTOR during abnormal ET-1 signaling under HG conditions provides protection against cellular oxidative damages.

Our above results showed that elevated glucose concentrations and abnormal ET-1 signaling induce significant levels of intracellular ROS and impaired mitochondrial function that acted as an important source of ROS generation. Therefore, we tested whether treatment with ERAs could mitigate the mitochondrial membrane potential (MMP) and measured the MMP across non-treated vs. treated cells. Supporting previous observations, we found that HG-induced cellular stresses also abrogated the MMP, which was synergistically altered with ET-1. However, when treated with inhibitors of either ET_A_-R (ambrisentan) or of both ET_A_-R and ET_B_-R (bosentan), the MMP impairment was significantly improved in HG-treated cells (Figure 2E). Altogether, these findings suggested that the HG condition leads to abnormally increased levels of ROS and involves a complex interplay between endothelin signaling and the mTOR pathway in H9c2 cells.

### 2.3. Inhibition of mTOR Alleviates HG-Induced Oxidative Stress and Improves the Survival Markers

To further assess the link between endothelin ET-1 signaling and mTOR pathway under HG conditions, we subjected the H9c2 cells to either HG treatment alone or to prior silenced Raptor or Rictor mTOR complex components and then treated cells with ET-1 and HG. Our results showed that the protein expression of survival markers p-Akt and Bcl-2 was significantly downregulated, while proapoptotic Bax expression was enhanced under HG treatment. This effect was further aggravated by treatment with ET-1, suggesting a contribution of ET-R signaling to HG-induced cardiomyocyte stresses. Interestingly, the inhibition of Raptor or Rictor effectively improved the survival markers and reduced the Bax levels (Figure 3A). Moreover, the protein expression of ROS-detoxifying catalase, heme oxygenase-1 (HO-1), and superoxide dismutase 2 (SOD2) was upregulated by combined treatment with HG and ET-1. This might reflect a cellular response to increased oxidative stress. Since the silencing of either Raptor or Rictor normalized the expression levels of antioxidant markers to the level of untreated cells, the mTOR pathway is likely to contribute to HG-induced intracellular stresses under the dictatorship of ET-1 signaling in H9c2 cells (Figure 3B). Furthermore, our results showed that HG-induced oxidative stress downregulated the protein expression level of mitochondrial biogenesis marker and redox regulator PGC1α, along with autolysosomal/autophagy regulator, transcription factor EB (TFEB), that was synergistically downregulated under HG plus ET-1 treatments. In contrast, genetic inhibition of Raptor or Rictor reduced this effect and TFEB and PGC1α expression was reverted to control levels (Figure 3C). Together, these results prove that endothelin signaling aggravates the HG-induced cellular stresses that suppress the expression of survival proteins. Upregulation of ROS-detoxifying enzymes under conditions of HG was not sufficient to counteract oxidative stress and the accompanying cellular damages. However, the inhibition of mTOR exhibited protection against impaired endothelin signaling under HG conditions in H9c2 cells.

### 2.4. Inhibition of mTOR Decreased ET-1 Mediated Mitochondrial Damage in HG-Treated H9c2 Cells

The above results showed that ET-1 signaling under HG conditions leads to significant intracellular oxidative stresses and mitochondrial damages involving the mTOR pathway. Therefore, we tested whether partial suppression of the mTOR pathway by inhibition of either Raptor or Rictor could mitigate the HG-induced mitochondrial damages. To further examine the mitochondrial damages, we measured the expression of mRNA markers associated with mitochondrial fragmentation/fission. We found that HG treatment leads to the upregulation of Drp-1 and Fis-1, which were more significantly increased when pretreated with ET-1, implicating that endothelin signaling under HG conditions increases mitochondrial fragmentation (Figure 4A). This observation was further confirmed by specifically examining the mitochondrial structures upon HG or HG plus ET-1 treatment and the silencing of either Raptor or Rictor. Our results showed that HG treatment leads to a significant decrease in mitochondrial interconnectivity (area/perimeter ratio), indicative of increased fragmentation/fission of mitochondria. This was paralleled with a decrease in mitochondrial tubule lengths suggestive of decreased mitochondrial elongation (form factor, 1/circularity). This mitochondrial structure impairment under HG treatment was further aggravated when pretreated with ET-1 (Figure 4B,C). However, the inhibition of either Raptor or Rictor significantly antagonized the adverse effects of ET-1 signaling under HG conditions. The mitochondrial morphology was reverted, as with the control cells upon the inhibition of mTOR under HG conditions in H9c2 cells (Figure 4B,C). Together, these results showed that HG-induced cellular stresses lead to the dysfunction of mitochondria, resulting in increased mitochondrial fragmentation and decreased mitochondrial size. This effect was further worsened when combined with endothelin ET-1. However, partial inhibition of mTOR (either Raptor or Rictor) abolished those adverse mitochondrial defects. These findings further implicated that ET-1 signaling under HG conditions leads to significant mitochondrial impairments involving contributions from the mTOR pathway, which could be amended by abrogation of either Raptor or Rictor.

## 3. Discussion

This study aimed to delineate the interplay between endothelin ET-1 signaling and the mTOR pathway under hyperglycemic conditions in H9c2 cardiomyoblasts and investigate whether treatment with ERAs or partial inhibition of mTOR could avert the HG-induced cellular damages. Cardiac oxidative stress varies depending on the balance between ROS production and detoxification [39]. SOD2, HO-1, and catalase impairment are critical factors in determining the cellular ROS levels that cause cardiac myocyte remodeling and failure [40]. In the cell culture system, glucose levels close to 10 mM are considered pre-diabetic levels, whereas glucose levels beyond 10 mM are mimicking diabetic condition [41]. This is crucial as proteins are altered by the processes of glycation and glycoxidation, leading to cellular damage, and eventually death under diabetic conditions. In this study, we determined that treatment with 30 mM glucose for 24 h leads to hyperglycemic injury, as already shown by several other studies [42]. Our dose-dependent analysis proved glucose concentrations of 30 mM and beyond lead to a significant loss of cell viability, with marked reduction in survival markers pAkt/Akt and Bcl-2, and an increase in cell death marker c-caspase 3. These results agreed with previous research, where high concentrations of glucose (22–33 mM) caused apoptotic cell death in cardiomyocytes [43,44]. Glucose transporters (GLUT1 and GLUT4) are known to be major glucose carriers across the plasma membrane in cardiomyocytes [45]. GLUT1 is primarily located on the plasma membrane and facilitates a major portion of the cardiac muscle’s resting glucose uptake. Contrarily, GLUT4 is primarily found in intracellular vesicles during the resting state and is transferred to the plasma membrane in response to insulin stimulation or environmental changes [46]. In healthy patients, insulin increases the translocation of GLUT4 to the plasma membrane through AMPK via activating PI3K and Akt [47,48], while studies in diabetic glomeruli have shown that the expression of GLUT1 and glucose flux both were increased as mTOR activity was increased and vice versa, involving a feed-forward mechanism [49,50].

In this study, our results showed that HG treatment upregulated the expression of GLUT1 and GLUT4 in H9c2 cells. Although the expression of insulin-dependent and insulin-independent GLUTs was increased, a state of glucose intolerance was induced in H9c2 cells, resulting in reduced glucose utilization, which could be significantly amended by insulin treatment. It has been demonstrated that ET-1 and ET_A_-Rs are elevated in an HG environment, which raises the risk of heart failure caused by diabetes. One of the main causes of ongoing diabetes-related damage in cardiomyocytes is HG-induced irreversible ET_A_-R overexpression. ET_A_-R activates phospholipase C (PLC), which, in turn, produces 1,4,5-inositoltriphosphate (IP_3_) that increases the release of Ca^2+^ from intracellular reserves. The impairment of endothelin signaling under diabetic conditions has been linked with a plethora of cardiac function abnormalities and heart failure [51,52]. ET-1 reduced the contractility of cardiomyocytes via inhibition of autophagy in the cardiac hypertrophic mice model [53]. Since it has been shown that altered intracellular Ca^2+^ homeostasis inhibits the formation of autophagosomes [54] and that ET-1 can activate Ca^2+^ mobilization pathways [38], measuring free Ca^2+^ and examining Ca^2+^ signaling after diabetic stimuli in isolated cardiomyocytes may be an important future strategy to study diabetes-associated autophagy impairment.

Of note, previous studies have shown that cardiac ET_A_-R ablation protects against aging-associated myocardial remodeling and contractile dysfunction through autophagy regulation. The study in H9c2 cells reported that the blocking of ET_A_-R using BQ123 or activating autophagy through an mTOR inhibitor rapamycin exhibited protection against adverse cardiac remodeling and contractile dysfunction through the correction of autophagy [13]. The present study further showed that HG condition resulted in increased expression of endothelin ET-1/ET_A_-R/ET_B_-R, together with mTOR/Raptor/Rictor that were paralleled with decreased expression of mitochondrial biogenesis marker, PGC1α and TFAM, in H9c2 cells. Studies by others have shown that mTOR, a crucial stress regulator, primarily encourages oxidative metabolism and mitochondrial biogenesis in a context dependent manner [55]. NADPH oxidase activation and mitochondrial dysfunction increased ROS generation through Akt-mTOR signaling [26]. It is known that mTOR plays a crucial role in maintaining oxidative equilibrium, is involved in the degradation of NRF2, and acts upstream of NRF2 in Ang II-induced cardiac hypertrophy [27,56]. The primary contributing cause to heart-related mortality in diabetes patients was discovered to be ROS-mediated cardiac problems [57]. This could be counteracted by the activity of antioxidant enzymes controlling ROS levels [58].

In line with previous studies, we demonstrated that HG causes increased ROS in H9c2 cells. We showed that H9c2 cells treated with a hyperglycemic dose of glucose generate intracellular ROS at a markedly increased rate. In cardiac cells under HG stress, our work demonstrated an increase in the expression of antioxidative enzymes SOD2, HO-1, and catalase. Although there was a compensatory increase in ROS defenses, it was insufficient to improve the toxic insults that resulted from increased ROS production. Noticeably, pretreatment with ET-1 further aggravated those HG-induced cellular injuries in H9c2 cells. Altogether, our results showed that the inhibition of either endothelin signaling by ablating ET_A_-R (ambrisentan) or blocking both ET_A_-R/ET_B_-R (bosentan) or partial suppression of mTOR complex components (Raptor/Rictor) significantly improved the HG-induced damages in H9c2 cells. The intracellular ROS level, as well as MMP, was improved in HG-treated cells. We also showed that HG-induced endothelin-mediated cellular stresses perturb the mitochondrial structures exhibited by the increased expression of fission genes Drp1 and Fis1. This study also showed that hyperglycemic condition with impaired endothelin signaling in H9c2 cells leads to increased mitochondrial fragmentation and decreased mitochondrial size. In addition, partial suppression of the mTOR complex improved the endothelin-mediated HG-induced mitochondrial damages. However, it should be noted that our findings derived from in vitro studies using H9c2 cardiomyoblasts. Thus, animal studies should be further performed to confirm the relationship between endothelin ET-1 signaling and mTOR pathways involved in HG-induced cardiomyocyte damage.

## 4. Materials and Methods

### 4.1. Materials

Endothelin-1 was obtained from Tocris Biosciences (Bristol, UK). Ambrisentan, bosentan, and 3-(4,5-dimethylthiazol-2yl)-2,5-diphenyl-2H-tetrazolium bromide (MTT) were obtained from Sigma-Aldrich (Saint Louis, MO, USA). Dulbecco’s Modified Eagle Medium (DMEM), phosphate-buffered saline (PBS), 0.25% trypsin-EDTA solution, fetal bovine serum (FBS), penicillin/streptomycin (P/S), and other cell culture reagents were acquired from Gibco (Grand Island, NY, USA).

### 4.2. Cell Culture

H9c2 rat cardiomyoblasts were acquired from the American Type Culture Collection (ATCC, CRL-1446). These cardiac cells were maintained in low glucose (5.5 mM) DMEM medium, supplemented with 10% FBS and 1% P/S solution at 37 °C in a CO_2_-buffered incubator. Upon attaining 80% confluence, cells were subcultured using 0.25% trypsin-EDTA solution to maintain the cell’s exponential growth stage and used for experimental treatments.

### 4.3. Cell Viability Assay

Cell viability assay was performed, as described earlier [59] with some modifications using MTT reagent, which is based on the transformation of tetrazolium salt MTT by active mitochondria to an insoluble formazan salt. Briefly, cells were seeded in 96-well plates (1 × 10^4^ cell/well) in low glucose DMEM medium and 1% P/S overnight. Cells were treated with various concentrations of D-Glucose (20–80 mM) for 24 h. After that, the medium was replaced with MTT solution (0.5 mg/mL), incubated for 4 h, and DMSO was added to dissolve the insoluble formazan product. The optical density (OD) was measured at λex 570 nm with a Clariostar microplate reader (BMG Labtech, Ortenberg, Germany). The results were calculated as a percentage of viable cells, compared with control cells [% cell viability = (absorbance of treated cells/absorbance of control cells) × 100].

### 4.4. Protein Expression Analysis by Western Blotting

The protein was harvested from cells, as described previously [59]. Briefly, after indicated treatment, cells were lysed in Triton X-100 lysis buffer (20 mM Tris pH 7.4, 0.8% Triton X-100, 150 mM NaCl, 2 mM EDTA, 10% glycerol, 100 μM phenylmethylsulfonyl fluoride, and protease inhibitor cocktail). After centrifugation, protein extracts were collected and quantified using Pierce™ BCA Protein Assay Kit (ThermoFisher, Waltham, MA, USA). Samples were mixed with 4X SDS Laemmli buffer and denatured by heating at 95 °C for 10 min. Next, samples were subjected to SDS-PAGE segregation and transferred to PVDF membrane (Bio-Rad, Hercules, CA, USA). Protein expressions were detected using specific primary antibodies: mTOR (Santa Cruz Biotechnology, Santa Cruz, CA, USA; sc-517464), p-mTOR (sc-293133), Raptor (sc-518004), Rictor (sc-271081), TFEB (sc-166736), PGC1α (sc-518025), Akt (Cell Signaling Technology, Danvers, MA, USA; cst#9272), p-Akt (cst#4060), Bax (cst#2772), Bcl-2 (cst#2876), c-caspase 3 (cst#9664), and GAPDH (Merck, Darmstadt, Germany; CB1001). The immunoblots were visualized using HRP-conjugated secondary antibody and chemiluminescent detection system iBright FL1500 Imaging System (ThermoFisher, Waltham, MA, USA). Protein densitometry analysis was done using Image J (NIH, Bethesda, MD, USA).

### 4.5. Measurement of Intracellular ROS Level

The intracellular ROS production was quantified using a fluorescent probe 2′,7′-dichlorodihydrofluorescein diacetate (DCFH-DA) (Sigma-Aldrich, St. Louis, MO, USA), as previously described [59]. The H9c2 cells were seeded in either a 96-well plate (1 × 10^4^ cell/well) or a 12-well plate (1 × 10^5^ cell/well) containing gelatin-coated coverslips overnight before treatment with various agents. After experimental treatments, the cells were rinsed with PBS. Next, 5 μM DCFH-DA was added, followed by incubation at 37 °C in the dark for 30 min. The fluorescent signal was captured using EVOS M7000 fluorescent imaging system (ThermoFisher, Waltham, MA, USA) or Clariostar microplate reader with wavelengths λex/λem: 485/530 nm.

### 4.6. Glucose Uptake Assay

The glucose uptake level was determined using Glucose Uptake Cell-Based Assay Kit (Cayman, Ann Arbor, MI, USA), as described previously [60]. The H9c2 cells were seeded in 96-well plates (1 × 10^4^ cell/well) in low glucose DMEM medium and 1% P/S overnight. Next, the cells were incubated without or with 30 mM D-Glucose-containing medium for 24 h. Then, the cells were incubated with fluorescently tagged glucose derivative (2-NBDG, 200 μg/mL) in the glucose-free culture medium for 30 min. Next, the cells were washed and treated with or without insulin (200 nM) for 30 min. Then, cells were rinsed with PBS, and 100 μL cell-based assay buffer was added to the cells. The fluorescence intensity was measured immediately using a Clariostar microplate reader at wavelengths λex/λem: 485/530 nm.

### 4.7. siRNA Transfection

The following siRNAs were procured from Santa Cruz Biotechnology: Raptor siRNA (sc-270140); Rictor siRNA (sc-270141). The H9c2 cells were transfected with either Raptor or Rictor siRNA (20 nM) or control siRNA using Lipofectamine reagent (ThermoFisher, Waltham, MA, USA), as described earlier [61] with some modifications. The siRNA duplex, as well as Lipofectamine 2000 reagent, was diluted with Opti-MEM (Gibco, Grand Island, NY, USA) and incubated for 5–10 min. Next, both solutions were mixed together, and incubated for 15–20 min. Subsequently, the above siRNAs mixture was added to the cells. All follow-up treatments were performed after 24 h of transfection.

### 4.8. Mitochondrial ROS Measurement

The mitochondrial superoxide production was assessed using a MitoSOX red (ThermoFisher, Waltham, MA, USA) reagent. Briefly, the H9c2 cells were seeded onto 96-well culture dish (1 × 10^4^ cells/well), as well as 12-well culture dish (1 × 10^5^ cells/well), containing gelatin-coated coverslips overnight before treatment with various agents in 1% FBS medium. After treatment, cells were rinsed with PBS and incubated with 5 μM MitoSOX at 37 °C for 20 min. Next, cells were washed with PBS, and the fluorescent signal was captured using EVOS M7000 fluorescent imaging system (ThermoFisher, Waltham, MA, USA) or Clariostar microplate reader with wavelengths λex/λem: 510/580 nm.

### 4.9. Quantitative Real-Time PCR (qPCR)

The total RNAs were extracted from H9c2 cells using GeneJET RNA Purification Kit (ThermoFisher, Waltham, MA, USA). RT-qPCR was performed on CFX96 Real-Time PCR Detection System (Bio-Rad, Hercules, CA, USA) using KAPA SYBR FAST One-step RT-qPCR kits (KAPA biosystems, Boston, MA, USA). Gene-specific primers for cell survival, mitochondrial markers, mTOR signaling, and ET-1 signaling were designed by Primer3 software. The list of primers used is provided in Appendix A. The expression level of target genes was normalized to reference gene GAPDH, and fold changes were calculated, according to the comparative cycle threshold (2^−ΔΔCT^) method.

### 4.10. Measurement of Mitochondrial Membrane Potential by JC-1 Staining

The mitochondrial membrane potential (MMP) was determined using JC-1 reagent (Abcam, Edmonton, AB, Canada), as per manufacturer instructions. The H9c2 cells were seeded in 96-well plates (1 × 10^4^ cell/well) in low glucose DMEM medium overnight before treatment with various agents in 1% FBS medium. In brief, after respective treatments, the cells were rinsed with PBS and incubated with a culture medium containing JC-1 reagent (2.5 μM) at 37 °C for 20 min. After incubation, the cells were rinsed with PBS twice, and fluorescence intensity was measured by a Clariostar microplate reader with wavelengths λex/λem: 590/530 nm.

### 4.11. Measurement Labelling

To stain mitochondria in living cells, MitoTracker Red CMXRos (ThermoFisher, Waltham, MA, USA), a red-fluorescent dye that stains mitochondria in live cells and its accumulation is dependent upon membrane potential, was used. The H9c2 cells were seeded onto 12-well culture dish (1 × 10^5^ cells/well) containing gelatin-coated coverslips overnight before treatment with various agents in 1% FBS medium. After treatment, cells were rinsed with PBS and incubated with 100 nM Mitotracker Red CMXRos at 37 °C for 20 min in serum-free medium. Next, cells were washed with PBS (x3), and images were captured using an ECLIPSE Ti2 confocal microscope (Nikon, Tokio, Japan) with 100X objective lens.

### 4.12. Statistical Analysis

Data are presented as mean ± SEM. All the results were representative of three or more independent experiments. The statistical analysis was performed using Student’s *t*-test and one-way analysis of variance (ANOVA) followed by Tukey post hoc test in GraphPad Prism (GraphPad Software, La Jolla, CA, USA). A value of *p* < 0.05 was considered statistically significant.

## 5. Conclusions

In this study, we showed that HG-induced cardiomyocyte damages involve increased endothelin signaling together with perturbed mTOR activity (Figure 5). To our knowledge, this is the first study implicating an interplay between ET-1 signaling and the mTOR pathway under hyperglycemic conditions in cardiomyocytes. Several questions are yet to be investigated, such as the role of auto-lysosomal regulator TFEB, mitochondrial dynamics, and the contribution of Ca^2+^ signaling in the complex interaction between endothelin signaling and mTOR under hyperglycemic conditions in cardiomyocytes. The interplay between ET-1 signaling and mTOR under hyperglycemic conditions in cardiomyocytes needs to be further investigated to unveil potential drug targets to treat diabetic cardiomyopathies.

## Figures and Tables

**Figure 1 ijms-23-13816-f001:**
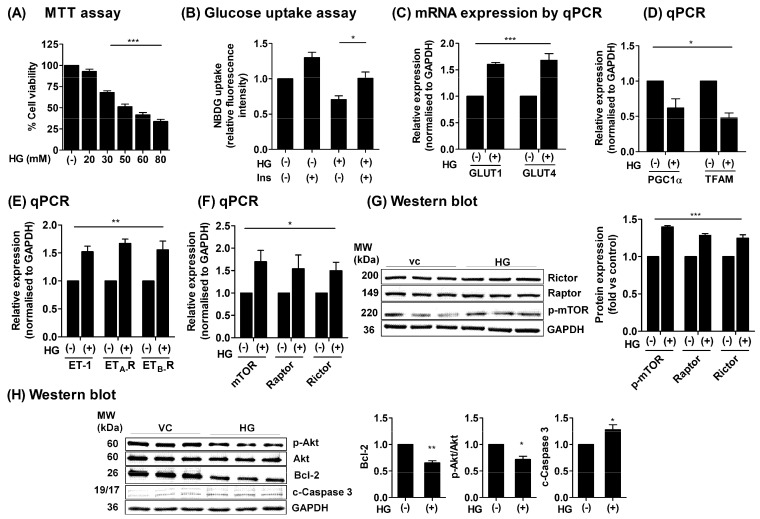
Effect of high glucose (HG) treatment in H9c2 cells. (**A**) The H9c2 cells were seeded in low glucose DMEM medium (5.5 mM) and treated with various concentrations of D-glucose-containing medium (vc = vehicle control, 20 mM, 30 mM, 50 mM, 60 mM, and 80 mM) for 24 h. Cell viability was assessed by MTT assay. (**B**) The H9c2 cells were seeded in low glucose DMEM medium overnight and incubated with HG medium for 24 h and subjected to 2-NBDG-based glucose uptake assay. The fluorescent intensity was measured at wavelengths λex/λem: 485/530 nm. (**C**–**F**) The H9c2 cells were treated with HG for 24 h, and total RNA was extracted. The mRNA expression level of GLUT1, GLUT4, ET-1, ET_A_-R, ET_B_-R, mTOR, Raptor, Rictor, PGC1α, and TFAM genes were determined by qPCR and normalized to GAPDH and expressed as fold versus non-treated controls (vc) (2^−ΔΔCT^). (**G**,**H**) The H9c2 cells were treated with HG as above, and proteins were extracted after 24 h of incubation. Protein expressions of p-mTOR/Raptor/Rictor (**G**) and survival markers p-Akt, Akt, Bcl-2, and c-Caspase-3 (**H**) were determined using western blotting. *N* = 3; Mean ± S.E.M.; * *p* < 0.05, ** *p* < 0.01, *** *p* < 0.001.

**Figure 2 ijms-23-13816-f002:**
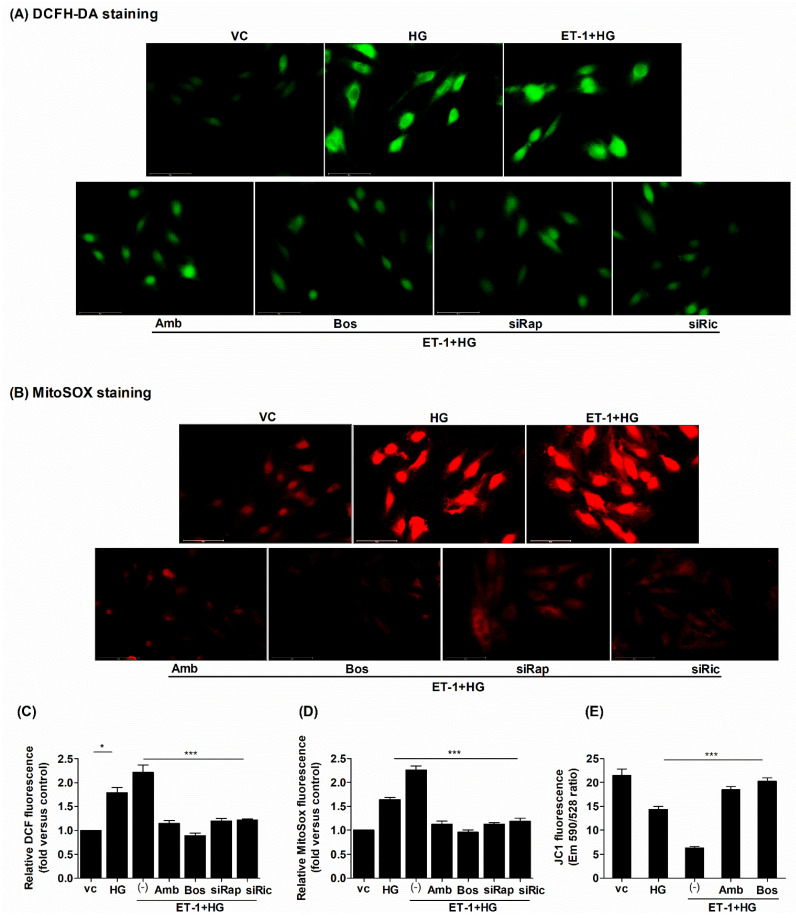
HG-induced oxidative damage was attenuated by ERAs or inhibition of mTOR. The H9c2 cells were seeded in low glucose medium overnight. Next, cells were treated with either HG alone for 24 h or pretreated with ambrisentan/bosentan (1 µM) for 1 h, followed by treatment with endothelin ET-1 (20 nM) for 3 h, followed by HG treatment for another 24 h. Additionally, another group of cells was transfected with siRNAs against Raptor/Rictor (20 nM) for 24 h followed by ET-1 (20 nM) treatment for 3 h and by HG treatment for another 24 h. (**A**,**C**) The intracellular ROS generation was determined using 5 μM DCFH-DA. The fluorescent signal was detected with wavelengths λex/λem: 485/530 nm. (**B**,**D**) MitoSOX (5 µM) was used to measure the mitochondrial superoxide production and the fluorescent intensity was captured at wavelengths λex/λem: 510/580 nm. (**E**) The MMP was estimated using JC-1 reagent (2.5 µM) and the fluorescent signal was recorded at wavelengths λem/λem: 590/530 nm. *N* = 4; Mean ± S.E.M.; * *p* < 0.05; *** *p* < 0.001.

**Figure 3 ijms-23-13816-f003:**
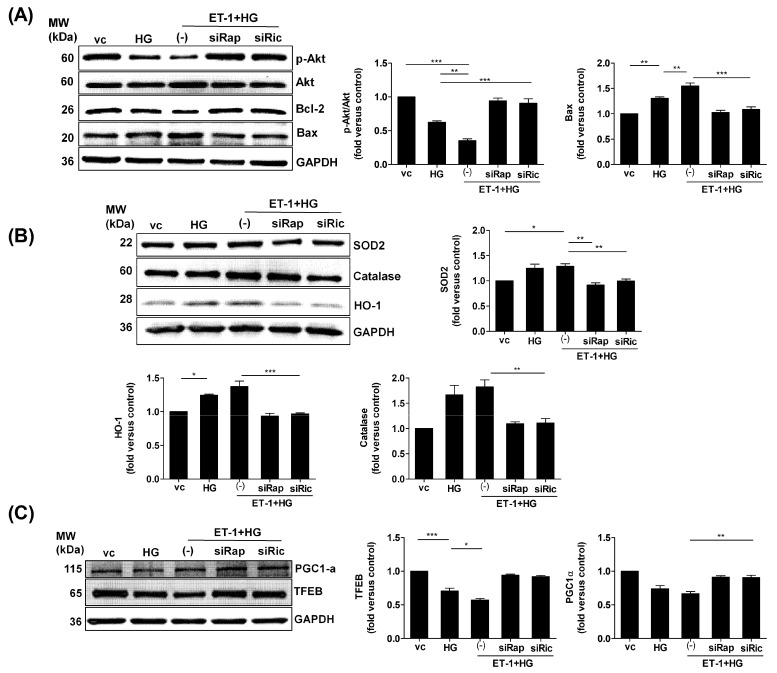
Inhibition of mTOR improved the survival markers in HG-treated H9c2 cells. The H9c2 cells were seeded in low glucose medium overnight. Next, the cells were transfected with siRNAs against Raptor/Rictor for 24 h, followed by ET-1 (20 nM) treatment for 3 h and by HG treatment for another 24 h. Protein expressions of survival/apoptosis markers p-Akt, Bcl-2, and Bax (**A**); antioxidant markers SOD2, catalase, and HO-1 (**B**); and mitochondrial/autophagy marker PGC1α and TFEB (**C**) were determined using western blot. Protein densitometry analysis was done using Image J software (NIH). *N* = 3; Mean ± S.E.M.; * *p* < 0.05, ** *p* < 0.01, *** *p* < 0.001.

**Figure 4 ijms-23-13816-f004:**
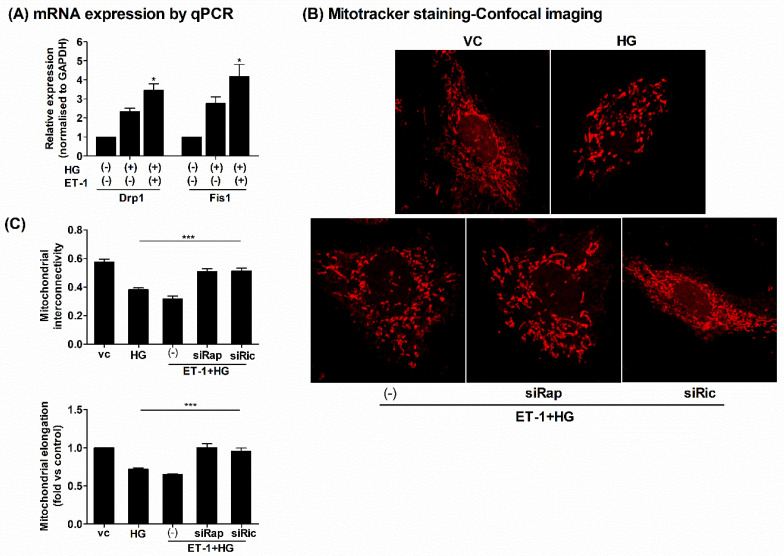
Inhibition of mTOR improved the mitochondrial damage under HG conditions in H9c2 cells. The H9c2 cells were seeded in 12-well culture dishes on gelatin-coated coverslips in low glucose DMEM medium. Next, the cells were transfected with siRNAs targeting Raptor/Rictor or control siRNAs for 24 h followed by ET-1 (20 nM) treatment for 3 h, followed by HG treatment for another 24 h. (**A**) The total RNAs were extracted, and q-PCR was performed to assess mRNA expressions of mitochondrial fission-related genes Drp1 and Fis1. The mRNA expressions were normalized to GAPDH and expressed as fold versus controls (vc) (2^−ΔΔCT^). (**B**) At the end of designated treatments, mitochondrial morphology was assessed, employing MitoTracker red labeling using confocal microscopy, and quantification is depicted in figure (**C**); the mitochondrial interconnectivity indicative of fragmentation/fission was calculated as area/perimeter ratio while mitochondrial elongation indicative of size was calculated as inverse of circularity and expressed as fold relative to control. At least 25–30 cells per group from three independent experiments were analyzed using Image J software (NIH). *N* = 3; Mean ± S.E.M.; * *p* < 0.05; *** *p* < 0.001.

**Figure 5 ijms-23-13816-f005:**
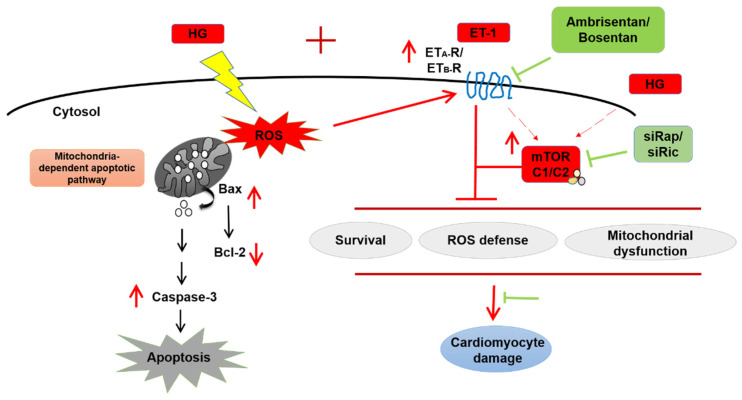
Schematic representing the interplay between ET-1/ET_A_/ET_B_ receptor and mTOR signaling under HG conditions in H9c2 cells. HG treatment leads to increased production of ROS, mitochondrial damage, decreased expression of survival markers, and increased expression of apoptosis markers. These HG-induced cardiomyocyte damages were ameliorated by inhibition of endothelin receptors ET_A_-R alone, both ET_A_-R/ET_B_-R, or partial suppression of mTOR complex. HG: High glucose; ROS: Reactive oxygen species; ET-1: Endothelin-1; siRap: silencing Raptor; siRic: silencing Rictor.

## Data Availability

All data generated or analyzed during the current study are included in this published article and its Appendix A.

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
