# Peer review of "High Glucose-Induced Cardiomyocyte Damage Involves Interplay between Endothelin ET-1/ETA/ETB Receptor and mTOR Pathway"

_ijms, 2022, doi:10.3390/ijms232213816_

Round 1

Reviewer 1 Report

This manuscript by Sudhir Pandey et al. have investigated the molecular crosstalk between endothelin ET-1 signaling and mTOR under HG conditions in H9c2 cells. they emphasized that ET-1 signaling under HG conditions leads to a significant mitochondrial dysfunction involving contributions from the mTOR pathway. While this information, in general, is of interest and add to the current literature illustrating the role of  endothelin-1 (ET-1) signaling and mTOR pathway in diabetic cardiomyopathy, there are several major critiques that have to be addressed by the authors and I have some concerns regarding the presentation, method design and interpretation of the data.

Major critiques:

  • The experimental strategies design and data reported are very poor and not very convincing as it based mainly on in vitro experiment and no data of in vivo study. It is difficult to understand exactly what type of diabetes in this study? Here they have mentioned diabetes mellitus but whether it is type 2 diabetes or type 1 diabetes is unclear.
  • If it is type 2 diabetes, why not type 1 diabetes? The authors mention type 2 diabetes in the majority of the manuscript, but type 1 diabetes in others; please unify the purpose of the study by focusing on one type of diabetes.
  • The materials and method section is very poor in terms of design and I am not at all convinced with this.
  • How many repeat experiments have been performed? At least one repeat experiment is required and has to be shown.

·         Technically one can criticise the use of only one standard gene during qRT-PCR which does not following MIQE-criteria that should be applied using qRT-PCR, same is true for western blotting.

·         Page 7 figure 3A: Although the phosphorylated proteins are normalized to GAPDH. It will be more accurate to also use respective total proteins (P AKT/ total AKT).

  • Please correct the manuscript for language and grammatical mistakes.

Reviewer 2 Report

Thank you very much for the opportunity to review an interesting work: High glucose-induced cardiomyocyte damage involves inter- 2 play between endothelin ET-1/ETA/ETB receptor and mTOR 3 pathway. The manuscript presents important research issues that will have an important application in practice due to the occurrence of type two diabetes mellitus.

The introduction is extensive and introduces the reader to the issue.

The results are presented cleary. Interestung and didactic fugures increase the value of manuscript.

Material and methods presents in detail the course of the study

Conclusions results presented clearly, especially Fig. 5

References 61 items are an overview of the new literature in the topic

I recommend the manuscript for further stages without changes

Author Response

We thank the reviewer for the insightful comments to our work.

Round 2

Reviewer 1 Report

Most of my concerns have been satisfactorily addressed by the authors. I therefore highly suggest that this work be published in IJMS.